# Neuronal Gaussian Process Regression

**Johannes Friedrich**
Center for Computational Neuroscience
Flatiron Institute
New York, NY 10010
jfriedrich@flatironinstitute.org

## Abstract

The brain takes uncertainty intrinsic to our world into account. For example, associating spatial locations with rewards requires to predict not only expected reward at new spatial locations but also its uncertainty to avoid catastrophic events and forage safely. A powerful and flexible framework for nonlinear regression that takes uncertainty into account in a principled Bayesian manner is Gaussian process (GP) regression. Here I propose that the brain implements GP regression and present neural networks (NNs) for it. First layer neurons, e.g. hippocampal place cells, have tuning curves that correspond to evaluations of the GP kernel. Output neurons explicitly and distinctively encode predictive mean and variance, as observed in orbitofrontal cortex (OFC) for the case of reward prediction. Because the weights of a NN implementing exact GP regression do not arise with biological plasticity rules, I present approximations to obtain local (anti-)Hebbian synaptic learning rules. The resulting neuronal network approximates the full GP well compared to popular sparse GP approximations and achieves comparable predictive performance.

## 1 Introduction

Predictive processing represents one of the fundamental principles of neural computations [1]. In the motor domain the brain employs predictive forward models [2], and a fundamental aspect of learned behavior is the ability to form associations between predictive environmental events and rewarding outcomes. These are just two examples of the general task of regression, to predict a dependent target variable given explanatory input variable(s), that the brain has to solve. The brain does not only predict point estimates but takes uncertainty into account, which led to coinage of the term "Bayesian brain" [3]. On the behavioral level, sensory and motor uncertainty have been shown to be integrated in a Bayesian optimal way [4]. There is also neurophysiological evidence, e.g. in the case of reward learning individual neurons in the orbitofrontal cortex (OFC) encode (average) value [5], while others explicitly encode the variance or 'risk' of the reward [6].

Above experimental findings lead to the corollary that the brain performs (non)linear regression while taking uncertainty into account. A principled framework to do so is a Gaussian process (GP) that has enjoyed prominent success in the machine learning community [7]. Furthermore, behavioral work in cognitive science suggests that people indeed use GPs for function learning [8, 9, 10]. In this paper I propose how the brain can implement (sparse) GP regression ((S)GPR).

**Contributions**   While the correspondence between infinitely wide Bayesian neural networks (NN) and GPs is well known [11], I show how the equations for the GP's predictive mean and variance can be mapped onto a specific NN of finite size. Further training its weights using standard deep learning techniques, it outperforms Probabilistic Back-propagation [12] and Monte Carlo Dropout [13].

Although a network wiring exists that exactly implements (S)GPR, it does not arise with biologically plausible plasticity rules. I present approximations to obtain local (anti-)Hebbian [14, 15] synaptic learning rules that result in a neuronal network with comparable performance as the exact NN.

**Biological evidence**    Tuning curves of my first layer neurons correspond to the GP kernel evaluated at training or inducing points for full or sparse GPs respectively. Prominent examples of neural tuning curves resembling (RBF) kernels are orientation tuning in visual cortex [16], place cells in hippocampus [17], and tuning curves in primary motor cortex [18]. Output neurons, e.g. in OFC for predicting reward, explicitly and distinctively encode predictive mean [5] and variance [6] of the encoded function evaluated at the current input. Synaptic learning rules are local and rely on prediction and risk prediction errors respectively, both of which have strong neurophysiological evidence [19, 20]. All other (hyper) parameters, such as the tuning curve centers [21], can be optimized using REINFORCE gradient estimates [22], which avoids biologically implausible error back-propagation. Using REINFORCE for biologically plausible updates was discussed by [23].

**Related work**    Several other works have investigated how the brain could implement Bayesian inference, cf. [24, 25] and references therein. They proposed neural codes for encoding probability distributions over one or few sensory input variables which are scalars or vectors, whereas a Gaussian process is a distribution over functions [7]. Earlier works considered neural representations of the uncertainty $p(\mathbf{x})$ of input variables $\mathbf{x}$, whereas this work considers the neural encoding of a probability distribution $p(f)$ over a dependent target function $f(\mathbf{x})$. To my knowledge, this is the first work to suggest how the brain could perform Bayesian nonparametric regression via GPs.

## 2   Background

In this section, I provide a brief summary of GPR and sparse GPR (SGPR) for efficient inference. I adopt the standard notation of [26], see Table S1 in the supplement for a summary of notation. I use boldface lowercase/uppercase letters for vectors/matrices and $\mathbf{I}$ for the identity matrix.

### 2.1   Gaussian process regression

Probabilistic regression is usually formulated as follows: given a training set of $n$ ($d$-dimensional) inputs $\mathbf{X} = \{\mathbf{x}_i\}_{i=1}^n$ and noisy (real, scalar) outputs $\mathbf{y} = \{y_i\}_{i=1}^n$, compute the predictive distribution of $y_*$ at test location $\mathbf{x}_*$. A standard regression model assumes $y_i = f(\mathbf{x}_i) + \epsilon_i$, where $f$ is an unknown latent function that is corrupted by Gaussian observation noise $\epsilon_i \sim \mathcal{N}(0, \sigma^2)$.

The GPR model places a (typically) zero-mean GP prior with covariance function $k(\mathbf{x}, \mathbf{x}')$ on $f$, i.e. any finite subset of latent variables follows a multivariate Gaussian distribution; in particular $p(\mathbf{f})^1 = \mathcal{N}(\mathbf{f}; 0, \mathbf{K}_{\mathbf{ff}})$ where $[\mathbf{K}_{\mathbf{ff}}]_{ij} = k(\mathbf{x}_i, \mathbf{x}_j)$. The covariance function $k(\mathbf{x}, \mathbf{x}')$ depends on hyperparameters, which are usually learned by maximizing the log marginal likelihood.

$$\log p(\mathbf{y}) = \log \mathcal{N}(\mathbf{y}; 0, \mathbf{K}_{\mathbf{ff}} + \sigma^2 \mathbf{I}) = -\tfrac{1}{2}\mathbf{y}^\top (\mathbf{K}_{\mathbf{ff}} + \sigma^2 \mathbf{I})^{-1}\mathbf{y} - \tfrac{1}{2}\log(|\mathbf{K}_{\mathbf{ff}} + \sigma^2 \mathbf{I}|) - \tfrac{n}{2}\log(2\pi). \quad (1)$$

In this simple model, the posterior over $f$, $p(f|\mathbf{y})$, can be computed analytically. The regression-based prediction for a test point $\mathbf{x}_*$ is a Gaussian distribution $p(y_*|\mathbf{y}) = \mathcal{N}(y_*; \mu_*, \Sigma_*)$. Introducing $\mathbf{k}_{\mathbf{f}*} = [k(\mathbf{x}_1, \mathbf{x}_*), ..., k(\mathbf{x}_n, \mathbf{x}_*)]^\top$ and $k_{**} = k(\mathbf{x}_*, \mathbf{x}_*)$, its predictive mean and variance are:

$$\mu_* = \mathbf{k}_{\mathbf{f}*}^\top (\mathbf{K}_{\mathbf{ff}} + \sigma^2 \mathbf{I})^{-1}\mathbf{y} \qquad (2)$$

$$\Sigma_* = k_{**} - \mathbf{k}_{\mathbf{f}*}^\top (\mathbf{K}_{\mathbf{ff}} + \sigma^2 \mathbf{I})^{-1}\mathbf{k}_{\mathbf{f}*} + \sigma^2 \qquad (3)$$

### 2.2   Sparse Gaussian process regression

The problem with the above expression is that inversion of the $n \times n$ matrix requires $\mathcal{O}(n^3)$ operations. This intractability can be handled by combining standard approximate inference methods with sparse approximations that summarize the full GP via $m \leq n$ inducing points leading to an $\mathcal{O}(nm^2)$ cost. A unifying view of early inducing point methods has been presented in [26], contemporary methods

have been unified in [27]. I focus on the popular sparse variational free energy (VFE) method [28], which performs approximate inference by maximizing a lower bound on the marginal likelihood of the data using a variational distribution $q(f)$ over the latent function [29]:

$$\log p(\mathbf{y}) \geq \log p(\mathbf{y}) - \mathrm{KL}[q(f)\|p(f|\mathbf{y})] = \log \mathcal{N}(\mathbf{y}; 0, \mathbf{Q_{ff}} + \sigma^2 \mathbf{I}) - \tfrac{1}{2\sigma^2}\mathrm{Tr}(\mathbf{K_{ff}} - \mathbf{Q_{ff}}) \quad (4)$$

where $\mathbf{Q_{ff}} = \mathbf{K_{fu}}\mathbf{K_{uu}^{-1}}\mathbf{K_{uf}}$ is the Nyström approximation of $\mathbf{K_{ff}}$ and $\mathbf{u}$ is a small set of $m \leq n$ inducing points at locations $\{\mathbf{z}_j\}_{j=1}^m$ so that $[\mathbf{K_{fu}}]_{ij} = k(\mathbf{x}_i, \mathbf{z}_j)$ and $[\mathbf{K_{uu}}]_{ij} = k(\mathbf{z}_i, \mathbf{z}_j)$. The first term corresponds to the deterministic training conditional (DTC, [26, 30]), the added regularization trace term prevents overfitting which plagues the generative model formulation of DTC. The prediction for a test point $\mathbf{x}_*$ is a Gaussian distribution $q(y_*) = \mathcal{N}(y_*; \mu_*, \Sigma_*)$ with predictive mean and variance:

$$\mu_* = \mathbf{k_{u*}^\top}(\mathbf{K_{uf}}\mathbf{K_{fu}} + \sigma^2 \mathbf{K_{uu}})^{-1}\mathbf{K_{uf}}\mathbf{y} \quad (5)$$

$$\Sigma_* = k_{**} - \mathbf{k_{u*}^\top}\mathbf{K_{uu}^{-1}}\mathbf{k_{u*}} + \mathbf{k_{u*}^\top}(\sigma^{-2}\mathbf{K_{uf}}\mathbf{K_{fu}} + \mathbf{K_{uu}})^{-1}\mathbf{k_{u*}} + \sigma^2 \quad (6)$$

## 3 Neural network representations for Gaussian process regression

By writing $\mu_* = \sum_i w_i k(\mathbf{x}_i, \mathbf{x}_*)$ where

$$\mathbf{w} = (\mathbf{K_{ff}} + \sigma^2 \mathbf{I})^{-1}\mathbf{y}, \quad (7)$$

we see that the mean prediction of a full GP in Eq. (2) is a linear combination of $n$ kernel functions, each one centered on a training point, which is one manifestation of the *representer theorem* [7]. For a sparse GP, cf. Eq. (5), it is a linear combination of $m$ kernel functions, each one centered on an inducing point $\mu_* = \sum_j w_j k(\mathbf{z}_j, \mathbf{x}_*)$ where

$$\mathbf{w} = (\mathbf{K_{uf}}\mathbf{K_{fu}} + \sigma^2 \mathbf{K_{uu}})^{-1}\mathbf{K_{uf}}\mathbf{y}. \quad (8)$$

Thus a simple linear neural network can implement the prediction of the mean. The neurons in the first layer correspond to inducing points. Their activities $\phi$ are kernel evaluations between a neuron's preferred stimulus, i.e. inducing point location (e.g. place cell center), $\mathbf{z}_j$ and the presented stimulus $\mathbf{x}_*$ (e.g. animal position), $\phi_j(\mathbf{x}_*) = k(\mathbf{z}_j, \mathbf{x}_*)$. The output layer consists of one (or more if predictions $y$ are not scalar but multidimensional) linear unit(s) with weights as defined above, cf. Fig. 1A. The mean prediction network has been called a regularization network in [31] because it was derived from the viewpoint of regularization theory, which is closely related to the maximum a posteriori probability (MAP) estimator in GP prediction, and thus omits uncertainty in predictions.

The term for the variance, Eq. (3) or Eq. (6), has the form $\Sigma_* = k_{**} + \sigma^2 - \mathbf{k_{u*}^\top}\mathbf{A}\mathbf{k_{u*}}$ with positive-definite matrix $\mathbf{A}$, and $\mathbf{u}$ replaced by $\mathbf{f}$ for a full GP. Decomposing $\mathbf{A}$ as $\mathbf{A} = \mathbf{U^\top U}$, e.g. using the Cholesky decomposition or the singular value decomposition, one obtains $\mathbf{k_{u*}^\top}\mathbf{A}\mathbf{k_{u*}} = (\mathbf{U}\mathbf{k_{u*}})^\top(\mathbf{U}\mathbf{k_{u*}}) = \sum_j (\mathbf{U}\mathbf{k_{u*}})_j^2 = \sum_j \psi_j$, where $\psi$ in the last equation is defined as $\psi_j = (\mathbf{U}\phi)_j^2$, which can be implemented in a 2-layer network, cf. Figs. 1A and S1. The neurons in the hidden layer have quadratic activation functions and are connected to the first layer with weights

$$\mathbf{U} = \left(\mathbf{K_{uu}^{-1}} - (\sigma^{-2}\mathbf{K_{uf}}\mathbf{K_{fu}} + \mathbf{K_{uu}})^{-1}\right)^{\frac{1}{2}}. \quad (9)$$

The output neuron has a linear activation function and sums up the activities of the hidden units. The additional term $k_{**} + \sigma^2$ merely adds a bias to the output neuron.

### 3.1 Learning

Thus far I derived an artificial neural network (ANN) that performs exact or sparse GPR. To obtain a biologically plausible neuronal network (BioNN) one needs to consider how the network connectivity, or at least an approximation to it, can arise with local synaptic learning rules. Throughout, I assume covariance functions that decay with distance, specifically I employ a squared exponential kernel with automatic relevance determination (ARD) $k(\mathbf{x}, \mathbf{x}') = s^2 \exp(-\frac{1}{2}\sum_{c=1}^d (x_c - x_c')^2/l_c^2)$.

I recognized that the analytic expression for $\mathbf{w}$ in Eq. (7) is the solution of a least squares problem with Lavrentiev regularization [32],

$$\mathbf{w} = \arg\min_{\tilde{\mathbf{w}}} \mathcal{L}(\tilde{\mathbf{w}}) \quad \text{with} \quad \mathcal{L}(\tilde{\mathbf{w}}) = \frac{1}{2}\|\mathbf{K_{ff}}\tilde{\mathbf{w}} - \mathbf{y}\|^2_{\mathbf{K_{ff}^{-1}}} + \frac{\sigma^2}{2}\|\tilde{\mathbf{w}}\|^2 \quad (10)$$

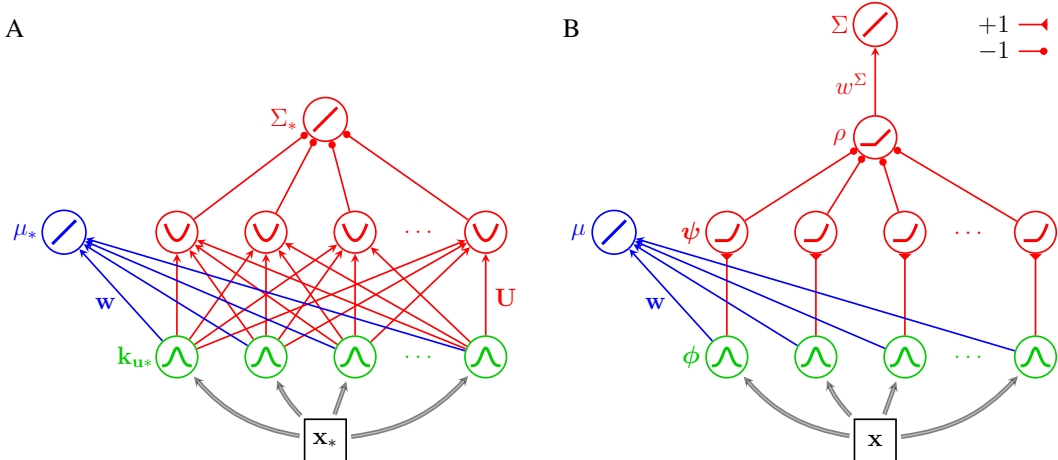

Figure 1: **A** Neural network for (S)GPR. The network outputs mean $\mu_*$ and variance $\Sigma_*$ of the predictive distribution for a test point $\mathbf{x}_*$. The neural activation functions are depicted within the nodes. Arrows are annotated with the synaptic weights. The special case of full GP regression is obtained for $\mathbf{u} = \mathbf{f}$ and $m = n$. **B** Biologically plausible neuronal network (BioNN) for SGPR. Plastic synapses are drawn as arrows. The weights of the static synapses are described in the legend. The linear output neurons could be replaced with linear-rectified units operating in the linear regime.

where I have used $\|\mathbf{x}\|_Q^2$ to stand for the weighted norm squared $\mathbf{x}^\top Q\mathbf{x}$. The gradient of the objective function $\mathcal{L}$ evaluates to $\frac{d\mathcal{L}}{d\mathbf{w}} = \mathbf{K_{ff}}\mathbf{w} - \mathbf{y} + \sigma^2\mathbf{w}$. However, in an online setting one only sees one data point $(\mathbf{x}_i, y_i)$ at a time and instead of gradient descent perform coordinate descent, $\Delta w_i = -\eta\left(\mathbf{w}^\top\mathbf{k_{f}}_i - y_i + \sigma^2 w_i\right)$ with learning rate $\eta$. Upon presentation of $(\mathbf{x}_i, y_i)$ only weight $w_i$ should be updated, but how do weights know which input has been presented? For a RBF kernel the presynaptic activity of neuron $i$ will be above, and the activity of all other neurons $j \neq i$ will be below a threshold, $\Theta(k_{ji} - s^2) = \delta_{ji}$, with Heaviside step function $\Theta(x) = 1$ if $x \geq 0$ else 0, and Kronecker delta $\delta_{ji} = 1$ if $j = i$ else 0. This yields the following synaptic learning rule for all neurons $j$ upon presentation of pattern $\mathbf{x}_i$:

$$\Delta w_j = -\eta \underbrace{\Theta\big(\phi_j(\mathbf{x}_i) - s^2\big)}_{\text{pre}}\underbrace{(\mathbf{w}^\top\boldsymbol{\phi}(\mathbf{x}_i) - y_i}_{\text{post}} \underbrace{+\, \sigma^2 w_j}_{\text{weight decay}}) \qquad \forall j. \qquad (11)$$

Importantly, the update involves merely a presynaptic input, a postsynaptic prediction-error ($\delta = \mu - y$), and a homeostatic term, that are all locally available to the synapse. However, the number of first layer neurons equals the number of data points $n$, i.e. a new neuron is recruited for every new data point. Hence, for even greater biological plausibility, I consider the case of SGPR where the number of first layer neurons is fixed to the number of inducing points $m$ in the remainder.

### 3.1.1 Predictive mean

I recognized that the analytic expression for $\mathbf{w}$ in Eq. (8) is the solution of a least squares problem with Tikhonov regularization [33],

$$\mathbf{w} = \arg\min_{\tilde{\mathbf{w}}} \mathcal{L}(\tilde{\mathbf{w}}) \quad \text{with} \quad \mathcal{L}(\tilde{\mathbf{w}}) = \frac{1}{2}\|\mathbf{K_{fu}}\tilde{\mathbf{w}} - \mathbf{y}\|^2 + \frac{\sigma^2}{2}\|\tilde{\mathbf{w}}\|_{\mathbf{K_{uu}}}^2. \qquad (12)$$

The gradient of the objective function $\mathcal{L}$ evaluates to $-\frac{d\mathcal{L}}{d\mathbf{w}} = \mathbf{K_{uf}}(\mathbf{y} - \mathbf{K_{fu}}\mathbf{w}) - \sigma^2\mathbf{K_{uu}}\mathbf{w} = \mathbb{E}_{i\sim\mathbf{f}}(n\mathbf{k_{u}}_i(y_i - \mathbf{w}^\top\mathbf{k_{u}}_i) - \sigma^2\mathbf{K_{uu}}\mathbf{w})$. The argument of the expectation is a gradient estimate to perform stochastic gradient descent in the biological setting of online learning. For covariances that decay with distance one can approximate $\mathbf{K_{uu}}$ by its diagonal $s^2\mathbf{I}$ to obtain a local learning rule:

$$\Delta w_j = -\eta\left(\underbrace{\phi_j(\mathbf{x}_i)}_{\text{pre}}\underbrace{(\mathbf{w}^\top\boldsymbol{\phi}(\mathbf{x}_i) - y_i)}_{\text{post}} + \underbrace{\frac{\sigma^2}{n}s^2 w_j}_{\text{weight decay}}\right) \qquad \forall j \qquad (13)$$

Indeed, well chosen inducing points tend to not cluster next to (or even on top of [34]) each other but to be well spread out over the entire data range, such that the off diagonal values are actually small.

Methods that have an exactly diagonal $\mathbf{K_{uu}}$ have been proposed [35], but these rely on spectral inter-domain features [36]. If $\sigma$ is small or $n$ large one can also neglect the noise term entirely.

### 3.1.2 Predictive variance

For the exact variance prediction one needs weights $\mathbf{U}$ given in Eq. (9). It is unclear to me how these weights can be learned in a biologically plausible manner, one can however approximate them. The second term in Eq. (9) is approximately zero and can be neglected compared to the first term, because $\sigma^{-2}\mathbf{k}_{\mathbf{f}j}^\top \mathbf{k}_{\mathbf{f}j} = \mathcal{O}(s^2 \frac{ns^2}{\sigma^2}) \gg k_{jj} = s^2$ as long as data size $n$ and signal-to-noise ratio $s/\sigma$ are not extremely small. One can approximate $\mathbf{K_{uu}}$ by its diagonal $s^2\mathbf{I}$, yielding weights $\mathbf{U} = s^{-1}\mathbf{I}$ that are constant, so no plasticity (rule) is necessary. Consequently, the input to the hidden layer neurons is always non-negative and the quadratic activation functions can be replaced with biologically realistic [37, 38, 39] half-squaring $\psi(\cdot) = (\max(\cdot, 0))^2$.

Thus far I assumed knowledge of the signal and noise level $s$ and $\sigma$ respectively. One can extend the neural net to estimate these quantities based on the data. I assume for now that the noise term in Eq. (8) is negligible and consider, without loss of generality, neural activations that are normalized to have a maximal activity of 1, i.e. $\phi_j(\mathbf{x}) = k(\mathbf{z}_j, \mathbf{x})/s^2$. Scaling $\phi$ by $s^{-2}$ merely results in weights $\mathbf{w}$ scaled by $s^2$, leaving the mean prediction $\mu_* = \mathbf{w}^\top \phi(\mathbf{x}_*)$ invariant. If one lets the weights $\mathbf{U}$ be identical to the identity matrix $\mathbf{U} = \mathbf{I}$, and the bias term be 1, then the output of the variance prediction network in Figs. 1A and S1 is the approximate non-normalized variance of $f_*$, $\rho(\mathbf{x}_*) \approx s^{-2}\mathbb{V}(f_*)$,[2] cf. Eq. (6) and Fig. S2. The variance of the observation $\mathbb{V}(y_*)$ is thus $s^2\rho(\mathbf{x}_*) + \sigma^2$, i.e. $\rho(\mathbf{x}_*)$ multiplied by some weight $w^\Sigma$ plus some bias $b^\Sigma$, and can therefore be represented by a linear neuron, cf. Fig. 1B. Weight and bias can be learned using a delta rule that minimizes the squared error between target value $\chi = \delta^2 = (y - \mu)^2$ and current prediction $\Sigma = w^\Sigma \rho + b^\Sigma$,

$$\Delta b^\Sigma = -\eta \left( w^\Sigma \rho + b^\Sigma - \chi \right) \tag{14}$$

$$\Delta w^\Sigma = -\eta \underbrace{\rho}_{\text{pre}} \underbrace{\left( w^\Sigma \rho + b^\Sigma - \chi \right)}_{\text{post}} \tag{15}$$

Importantly, the update involves merely a presynaptic and a postsynaptic term, that are all locally available to the synapse. To provide the target value $\chi$ one merely needs to introduce a neuron with quadratic activation function, which might rather be encoded by two complementary half-squaring neurons [38], that takes $\delta$ as input from the mean prediction network, cf. Fig. S2. Neurons encoding the postsynaptic 'risk prediction error' term $\Sigma - \chi$ have been reported in OFC [20].

Once learning converged the weight encodes the signal strength $w^\Sigma = s^2$ and the bias the noise level $b^\Sigma = \sigma^2$. These values can be read out in form of neural activity, if one assumes "up" and "down" states in the cortex [40] implement on and off switching of the bias respectively. Transitioning from "up" to "down" state the network output switches from variance $\mathbb{V}(y_*)$ to $\mathbb{V}(f_*)$. If no input $\mathbf{x}_*$ is provided, i.e. $\phi = 0$ and $\rho = 1$, the activity of the output neuron is the signal strength $s^2$ in the "down" state and the sum of signal strength $s^2$ and noise variance $\sigma^2$ in the "up" state.

### 3.1.3 Receptive field plasticity

Until now I assumed that the positions $\{\mathbf{z}_j\}_{j=1}^m$ of the inducing points, i.e. tuning curve centers, are given. While regular equidistant or even random placements (e.g. approximate determinant based sampling, [35]) can be quite effective, the locations can also be optimized. Such tuning curve adaptation is also observed experimentally [21, 41].

The usual approach is to follow the gradient of some objective function $\mathcal{L}$, e.g. the objective function in Eq. (12) or the ELBO in Eq. (4) that is maximized in the VFE method. The gradient of $\mathcal{L}$ with respect to $\mathbf{z}_j$ is obtained using the chain rule as the product of $\frac{d\mathcal{L}}{dk_{ij}}$ (where $k_{ij} = [\mathbf{K_{fu}}]_{ij}$) and the derivative of the activation function $\frac{dk_{ij}}{d\mathbf{z}_j} = k_{ij}(\mathbf{x}_i - \mathbf{z}_j)/l^2$. Updating the tuning curve of neuron $j$ would thus require not only knowledge of its own activity $k_{ij} = \phi_j(\mathbf{x}_i)$ and the difference between presented and preferred stimulus $(\mathbf{x}_i - \mathbf{z}_j)$, but also $\frac{d\mathcal{L}}{dk_{ij}}$, which is questionable from a biological point

of view. While the same conclusion holds for more complex objectives such as the ELBO, I consider for simplicity the gradient of the data fit term in Eq. (12). $\frac{d}{dk_{ij}} \frac{1}{2}\|\mathbf{K_{fu}}\mathbf{w} - \mathbf{y}\|^2 = (\mathbf{w}^\top \boldsymbol{\phi}(\mathbf{x}_i) - y_i)w_j$ would require either implausible symmetric feedforward and feedback connections to back-propagate the error, or a global error signal $(\mathbf{w}^\top \boldsymbol{\phi}(\mathbf{x}_i) - y_i)$ and knowledge of efferent synaptic strength $w_j$. The global error signal could well be encoded by neuromodulatory signals such as dopamine, but because synaptic strength depends on postsynaptic quantities such as dendritic spine size and number of receptors a neuron does likely not know its efferent synaptic efficacies.

Instead I suggest to perform updates using the (unbiased) gradient estimates of REINFORCE [22]. In order to minimize some objective function $\mathcal{L}(\{\mathbf{z}_j\})$ the $\mathbf{z}$ are perturbed $\mathbf{z}' = \mathbf{z} + \boldsymbol{\xi}$ with $\boldsymbol{\xi} \sim P(\boldsymbol{\xi})$. For the gradient of the expectation $\langle \mathcal{L} \rangle$ holds $\nabla_{\mathbf{z}} \langle \mathcal{L} \rangle = \langle (\mathcal{L} - B)\nabla_{\mathbf{z}} \log P(\mathbf{z}') \rangle$ where baseline $B$ is some (optional) control variate. For Gaussian distributed $\boldsymbol{\xi} \sim \mathcal{N}(0, \epsilon^2 \mathbf{I})$ the so called 'characteristic eligibility' or 'score function' is $\nabla_{\mathbf{z}} \log P(\mathbf{z}') = \boldsymbol{\xi}/\epsilon^2$, yielding the simple update rule

$$\Delta \mathbf{z}_j = -\eta \underbrace{(\mathcal{L} - B)}_{\text{global modulatory signal}} \underbrace{(\mathbf{z}'_j - \mathbf{z}_j)}_{\text{perturbation } \boldsymbol{\xi}_j} \qquad \forall j \qquad (16)$$

where the objective $\mathcal{L}$ is for example the squared prediction error that I already introduced earlier

$$\mathcal{L} = \chi = \delta^2 = ((k(\mathbf{z}'_1, \mathbf{x}_i), ..., k(\mathbf{z}'_m, \mathbf{x}_i))\mathbf{w} - y_i)^2 . \qquad (17)$$

The same method can be used to not only update the centers of the tuning curves, but also their widths $l$. Whereas a GP kernel uses one length scale (or $d$ for an ARD kernel and $d$-dimensional input), it seems far fetched to assume that the tuning curves of all neurons vary in a coordinated way. Therefore I let each neuron have its own length scale $l_j$ and update it analogously to Eq. (16). The additional flexibility of varying widths for basis functions further permits better sparse approximations [42].

Taken together we are thus equipped with methods to update all hyperparameters.

## 4  Experiments

I compare my NNs with GP implementations of GPy [43]. (My source code can be found at `https://github.com/j-friedrich/neuronalGPR`). The performances are compared using two metrics: root mean square error (RMSE) and negative log predictive density (NLPD).

I applied my derived neuronal networks to the Snelson dataset [44], that has been widely used for SGPR. Throughout I considered ten 50:50 train/test splits. I first studied how the synaptic weights can be learned online by performing the synaptic plasticity update for each presented data pair $(\mathbf{x}_i, y_i)$, passing multiple times over the training data. (Fig. S5 considers a streaming case that does not revisit data points.) Fig. 2A shows the RMSE for the test data while learning the weights to represent the mean of a full GP using Eq. (11). The hyperparameters were set to the values obtained with GPy. After few epochs the weights converged, yielding the same performance as the GP. Fig. 2B and C shows the RMSE and NLPD while learning the weights to represent the mean and variance of a sparse GP with $m = 6$ inducing inputs using the BioNN depicted in Fig. 1B with learning rules Eqs. (13-15). The length scale of the kernel and the positions of inducing points were set to the values obtained with VFE throughout the paper until mentioned otherwise later. The other hyperparameters $(s, \sigma)$ are automatically inferred by the network and the noise term in Eq. (13) has been neglected. Although my network has been designed to approximate the predictive distribution of VFE, the network converges to RMSE and NLPD values that outperform the VFE result. I attribute that to two facts. First, VFE tends to over-estimate the noise [34], cf. Fig. 3. Second, my network considers the output variable $\mathbf{y}$ to calibrate the noise, whereas the predictive variance of VFE (and full GP) only takes the inputs $\mathbf{X}$ into account.

In my network derivation I alluded to negligible noise terms and well separated inducing inputs that render $\mathbf{K_{uu}}$ close to diagonal. It is therefore of interest to study the influence of noise variance $\sigma^2$ and number of inducing points $m$. I considered ten 50:50 train/test splits. Fig. 3A depicts fits for full GP, VFE and BioNN (with converged weights) for one split. In Fig. 3B and C I scaled the noise in the data up and down by two orders of magnitude. My network predicts the noise variance more accurately than VFE and FITC [26, 44] and performs better according to NLPD. I confirm the finding reported in [34] that VFE tends to over- and FITC to under-estimate the noise. This is also visible in Fig. 3D where I varied the number of inducing points. Fig. 3E shows that the predictive

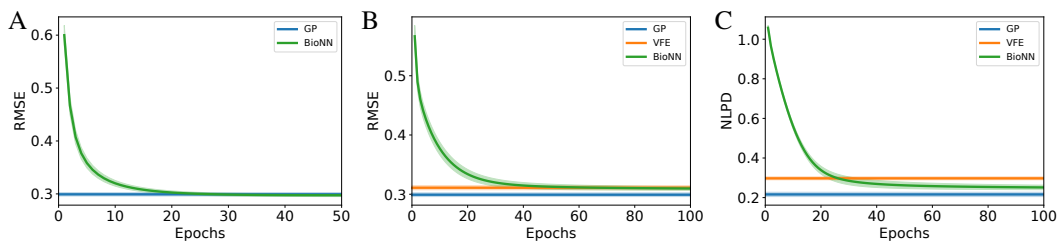

Figure 2: Online learning the weights of biologically plausible NNs for the Snelson dataset [44]. **A** Root mean square error (RMSE) for GPR trained with coordinate descent, Eq. (11). Lines and shaded areas depict mean $\pm$ SEM. **B** RMSE and **C** Negative log predictive density for SGPR trained with stochastic gradient descent, Eqs. (13-15).

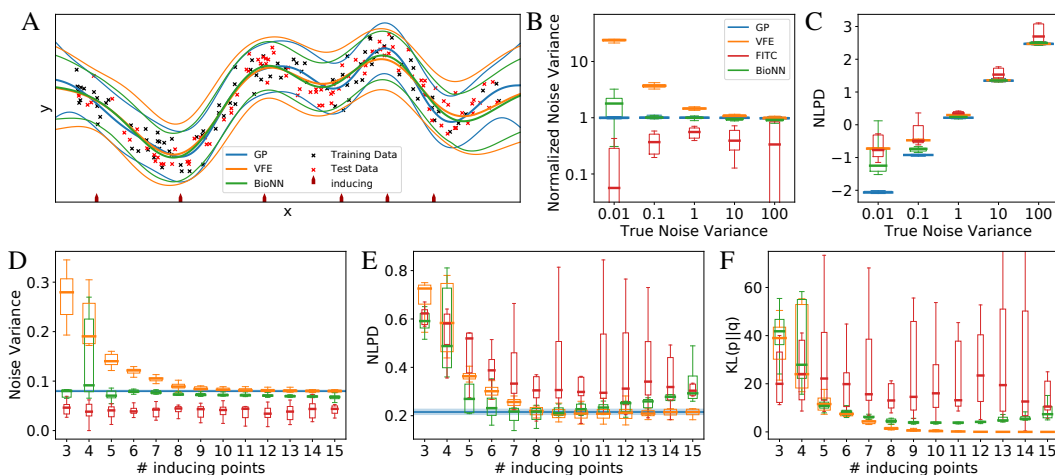

Figure 3: Comparison of my BioNN for SGPR, cf. Fig. 1B, with full GP, VFE and FITC. **A** Fits for full GP, VFE and BioNN. Thick and thin lines represent mean and 95% confidence region (mean $\pm$ 2 standard deviations) respectively. **B** Tukey boxplots of inferred noise variance normalized by true noise variance, and **C** NLPD as function of the noise level. **D** Tukey boxplots of inferred noise variance, **E** NLPD, and **F** KL divergence between full GP $p$ and sparse approximation $q$ as function of the number of inducing inputs.

performance of my network is comparable to VFE and FITC, and for about 6 to 10 neurons even to the full GP. Only for an unnecessary large, and metabolically costly, number of neurons does the diagonal approximation of $\mathbf{K_{uu}}$ break down, whereas VFE never worsens when adding inputs. Although we are mostly interested in good predictive performance, I also evaluated how well my BioNN approximates the GP. Fig. 3F shows that it does not do quite as well as VFE but better than FITC, as measured by the divergence $\mathrm{KL}(p(\mathbf{y}_*|\mathbf{y})\|q(\mathbf{y}_*))$ between the true test posterior $p(\mathbf{y}_*|\mathbf{y})$ and each of the approximate test posteriors.

I next evaluated the performance of my BioNN on larger and higher dimensional data. I replicate the experiment set-up in [12] and compare to the predictive log-likelihood of Probabilistic Back-propagation [12] and Monte Carlo Dropout [13] on ten UCI datasets [45], cf. Table 1. I set the number of inducing points equal to the number of hidden layer neurons in [12, 13]. For the too big *Year Prediction MSD* dataset I used the Stochastic Variational GP of [46]. Again, the kernel length scales and the inducing point positions of the BioNN were set to the values obtained with VFE. On these tasks VFE performs about as well as, if not better than, Dropout and PBP. Fig. 4 reveals overall comparable performance of my BioNN to VFE and FITC. (As a biologically plausible control baseline, I also considered a RBF network that connects not only the mean but also the variance predicting neuron directly to the first layer neurons, but it performed badly due to overfitting.) Although the main objective is good predictive performance, I was also interested in how well my BioNN approximates the GP. For the five datasets with merely $\mathcal{O}(1,000)$ data points I was able to fit the full GP. Table 2 shows that my BioNN approximates the full GP nearly as well as VFE and much

Table 1: Characteristics of the analyzed data sets, and average predictive log likelihood ± Std. Errors for Monte Carlo Dropout (Dropout, [13]), Probabilistic Back-propagation (PBP, [12]), sparse GP (VFE, [28]), an artificial neural network (ANN) with architecture corresponding to a sparse GP (but differing weights), cf. Fig. 1A, and a biologically plausible neural network (BioNN), cf. Fig. 1B.

| Dataset | $n$ | $d$ | Dropout | PBP | VFE | ANN | BioNN |
|---|---|---|---|---|---|---|---|
| Boston Housing | 506 | 13 | -2.46±0.06 | -2.574±0.089 | -2.483±0.050 | **-2.424±0.060** | -2.605±0.087 |
| Concrete Strength | 1,030 | 8 | **-3.04±0.02** | -3.161±0.019 | -3.161±0.016 | -3.089±0.025 | -3.180±0.026 |
| Energy Efficiency | 768 | 8 | -1.99±0.02 | -2.042±0.019 | -0.712±0.025 | **-0.660±0.032** | -0.729±0.038 |
| Kin8nm | 8,192 | 8 | 0.95±0.01 | 0.896±0.006 | 0.972±0.003 | **1.058±0.005** | 1.031±0.005 |
| Naval Propulsion | 11,934 | 16 | 3.80±0.01 | 3.731±0.006 | 8.800±0.022 | **9.171±0.012** | 9.059±0.015 |
| Power Plant | 9,568 | 4 | -2.80±0.01 | -2.837±0.009 | -2.810±0.009 | **-2.796±0.014** | -2.807±0.010 |
| Protein Structure | 45,730 | 9 | -2.89±0.00 | -2.973±0.003 | -2.894±0.005 | **-2.809±0.008** | -2.887±0.006 |
| Wine Quality Red | 1,599 | 11 | **-0.93±0.01** | -0.968±0.014 | -0.957±0.013 | -0.938±0.014 | -0.978±0.016 |
| Yacht Hydrodynamics | 308 | 6 | -1.55±0.03 | -1.634±0.016 | -0.717±0.041 | **0.060±0.042** | -0.867±0.102 |
| Year Prediction MSD | 515,345 | 90 | -3.59± NA | -3.603± NA | -3.613± NA | **-3.430± NA** | -3.612± NA |

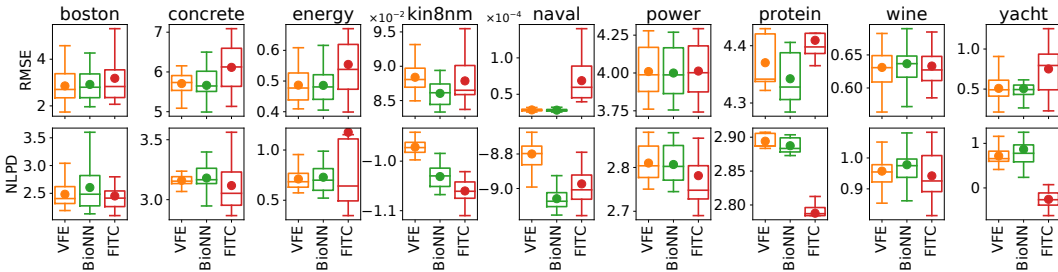

Figure 4: Comparison of my BioNN for SGPR, cf. Fig. 1B, with VFE and FITC on nine UCI datasets using $M = 50$ (100 for the protein dataset) inducing points. Shown are Tukey box plots as well as the means (filled circles).

better than FITC, as measured by the KL divergence between the true and each of the approximate test posteriors.

Taking a little detour and ignoring biological plausibility for a short moment, I was interested in how the ANN, Fig. 1A, performs when its weights are only initialized to values that correspond to a sparse GP, Eqs. (8-9), but then trained using standard deep learning techniques. I used the negative log-likelihood as loss function and performed 40 passes over the available training data using the Adam optimizer [47] with learning rate tuned by splitting the training data into a new 80:20 train/validation split. As Table 1 shows the ANN outperforms Dropout and PBP on almost all datasets. For deeper networks the quality of Dropout (and PBP) predictions increases slightly, but even then my ANN remains competitive and does not suffer from long prediction times needed to draw multiple MC samples (Fig. S3).

Thus far I set the tuning curves to be the VFE kernels at the inducing points, and were primarily interested in how the approximations needed to render the network biologically plausible affect the performance compared to VFE. Fig. 5 shows how the centers, as well as the widths, of the tuning curves can be learned using REINFORCE, Eq. (16). For each train/test split the 6 tuning curve centers were initialized on a regular grid at {0.5, 1.5, ..., 5.5} and updated to minimize the squared prediction error. As control variate I used a running average of the MSE. This resulted in predictions on the test

Table 2: Average KL($p\|q$) and Std. Errors between full GP $p$ and sparse approximation $q$.

| Dataset | VFE | BioNN | FITC |
|---|---|---|---|
| Boston Housing | 15.37 ± 0.83 | 29.05 ± 1.54 | 527.89 ± 64.73 |
| Concrete Strength | 25.64 ± 1.38 | 29.71 ± 2.04 | 69,425.31 ± 12,791.64 |
| Energy Efficiency | 4.79 ± 1.74 | 5.24 ± 1.88 | 103,854.63 ± 42,970.59 |
| Wine Quality Red | 482.06 ± 16.32 | 494.57 ± 16.21 | 5,391.73 ± 941.12 |
| Yacht Hydrodynamics | 26.92 ± 1.47 | 31.96 ± 3.00 | 2,721.97 ± 248.67 |

data that outperform VFE, and possibly even full GP. When applied to the annular water maze task [48] optimizing inducing points leads to the experimentally observed place cell accumulation effect at the goal location [48, 49] (Fig. S4).

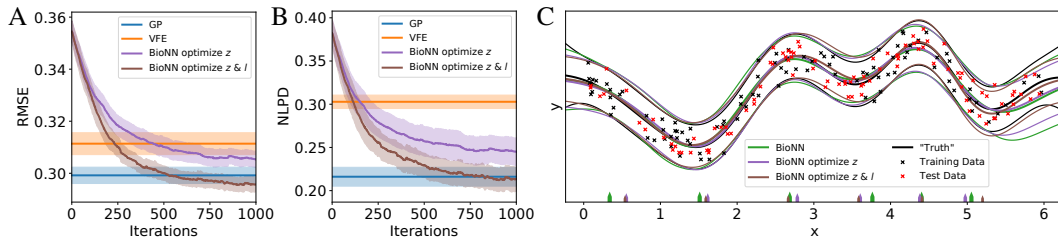

Figure 5: Tuning curve adaptation using REINFORCE [22]. **A** RMSE and **B** NLPD decrease with iterations. Lines and shaded areas depict mean $\pm$ SEM. **C** Fits for BioNN with tuning curves from VFE (green), optimized tuning curve centers $z$ (purple), and optimized tuning curve centers $z$ and width $l$ (brown). Black lines show the "Truth" obtained with a full GP on the combined test and training data.

## 5    Conclusion

I have introduced a biologically plausible Gaussian process approximation with good predictive performance and close approximation of the full Gaussian process. As real world regression example I considered the case of associating spatial locations with rewards. Once simultaneous recordings of e.g. place cells in hippocampus, the first layer of my network, and reward and risk prediction cells in OFC [50, 51] (potentially also ventral striatum [52, 53, 54]), the output layer of my network, in freely behaving animals become feasible, a more direct test of my predictions will be possible.

## Broader Impact

This paper introduces a biologically plausible implementation of Gaussian processes. It bridges the fields of machine learning and neuroscience with potential impact in both fields. With regard to machine learning this paper shows a correspondence between Gaussian processes and certain neural networks (of finite size) and raises the question of how best to perform nonlinear regression with uncertainty estimates. Should one use Gaussian processes, neural networks, or a combination of both – such as the presented Gaussian process initialized neural networks? With regard to neuroscience the paper introduces a biologically plausible Gaussian process approximation with good predictive performance and close approximation of the full Gaussian process, compared to VFE and FITC. It yields initial results in line with existing experimental data and motivates new experiments for a more direct test of the model.

Ethical aspects and future societal consequences do not apply to this work.

## Acknowledgments and Disclosure of Funding

The author was internally funded by and received his salary from the Simons Foundation (https://simonsfoundation.org). The author has declared that no competing interests exist.

## Footnotes

[1]Here I have collected the latent function values into a vector $\mathbf{f} = \{f(\mathbf{x}_i)\}_{i=1}^n$. The dependence on the inputs $\{\mathbf{x}_i\}_{i=1}^n$ and hyperparameters is suppressed throughout to lighten the notation.

[2] $\rho(\mathbf{x}_*) = \lfloor -\mathbf{1}^\top \psi(\mathbf{x}_*) + 1 \rfloor_+ = \lfloor 1 - \phi(\mathbf{x}_*)^\top \phi(\mathbf{x}_*) \rfloor_+ = \lfloor s^{-2}(k_{**} - \mathbf{k}_{\mathbf{u}*}^\top \operatorname{diag}(\mathbf{K}_{\mathbf{uu}}^{-1})\mathbf{k}_{\mathbf{u}*}) \rfloor_+ \approx s^{-2}\mathbb{V}(f_*)$. Here I added linear rectification $\lfloor \cdot \rfloor_+ = \max(\cdot, 0)$ to ensure that my approximations do not result in negative variance estimates.

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
