[Supplementary Material]

# Neuronal Gaussian Process Regression
# – Supplementary Material –

**Johannes Friedrich**
Center for Computational Neuroscience
Flatiron Institute
New York, NY 10010
jfriedrich@flatironinstitute.org

Table S1: Summary of notation.

| Notation | Description |
| --- | --- |
| KL | Kullback-Leibler divergence |
| Tr | matrix trace |
| $\mathbb{E}(X)$ | expected value of $X$ |
| $\mathbb{V}(X)$ | variance of $X$ |
| $\mathcal{N}(\mathbf{y}; \mathbf{m}, \mathbf{K})$ or $\mathbf{y} \sim \mathcal{N}(\mathbf{m}, \mathbf{K})$ | $\mathbf{y}$ is normally distributed with mean $\mathbf{m}$ and covariance $\mathbf{K}$, $p(\mathbf{y}) = \frac{e^{-\frac{1}{2}(\mathbf{y}-\mathbf{m})^\top \mathbf{K}^{-1}(\mathbf{y}-\mathbf{m})}}{\sqrt{|2\pi\mathbf{K}|}}$ |
| $\mathbf{I}$ | identity matrix |
| $\mathbf{1} = (1, ..., 1)^\top$ | vector of ones |
| $n$ | number of training data points |
| $d$ | dimensionality of inputs |
| $m$ | number of inducing points / neurons in first layer |
| $f$ | latent function |
| $k(\mathbf{x}, \mathbf{x}')$ | covariance function |
| $\sigma^2$ | noise variance |
| $s^2$ | signal variance |
| $\{l_c\}_{c=1}^d$ | length-scales / width of tuning curves along each dimension |
| $\mathbf{X} = \{\mathbf{x}_i\}_{i=1}^n$ | ($d$-dimensional) training inputs |
| $\mathbf{y} = \{y_i\}_{i=1}^n$ | (real, scalar) training outputs |
| $\mathbf{f} = \{f(\mathbf{x}_i)\}_{i=1}^n$ | latent function values at input points |
| $\mathbf{x}_*$ | test point |
| $\mathbf{Z} = \{\mathbf{z}_j\}_{j=1}^m$ | inducing point locations / tuning curve centers |
| $\mathbf{K_{ff}}$ | covariance matrix at input locations $\mathbf{X}$ |
| $\mathbf{K_{uu}}$ | covariance matrix at inducing point locations $\mathbf{Z}$ |
| $\mathbf{K_{fu}} = \mathbf{K_{uf}^\top}$ | covariance matrix between input locations $\mathbf{X}$ and inducing point locations $\mathbf{Z}$ |
| $\mathbf{k_{f*}}$ | covariance vector between input locations $\mathbf{X}$ and test point $\mathbf{x}_*$ |
| $\mathbf{k_{u*}}$ | covariance vector between inducing point locations $\mathbf{Z}$ and test point $\mathbf{x}_*$ |
| $\mathbf{Q_{ff}} = \mathbf{K_{fu}}\mathbf{K_{uu}^{-1}}\mathbf{K_{uf}}$ | Nyström approximation of $\mathbf{K_{ff}}$ |
| $\mu_*, \Sigma_*$ | predictive mean and variance for test point $\mathbf{x}_*$ / activity of the two output neurons |
| $\boldsymbol{\phi}(\cdot) = \{\phi_j(\cdot)\}_{j=1}^m = \{k(\mathbf{z}_j, \cdot)\}_{j=1}^m$ | tuning curves / activations of 1st layer neurons |
| $\boldsymbol{\psi}(\cdot) = \{\psi_j(\cdot)\}_{j=1}^m$ | activations of 2nd layer neurons |
| $\mathbf{w}, \mathbf{U}, w^\Sigma$ | synaptic weights |
| $b^\Sigma$ | bias |
| $\eta$ | learning rate |
| $\delta = \mu - y$ | prediction error |
| $\chi = \delta^2$ | squared prediction error |
| $\rho(\mathbf{x}_*)$ | non-normalized variance of $f_*$ / activity of 3rd layer neuron in Fig. S2 |
| $\boldsymbol{\xi}$ | perturbation of inducing point location / tuning curve center |
| $B$ | baseline, control variate to reduce variance of the gradient estimate |

Figure S1: Neural network for GP regression. The network outputs mean $\mathbb{E}(y_*)$ and variance $\mathbb{V}(y_*)$ of the predictive distribution for a test point $\mathbf{x}_*$. The nodes and arrows are annotated with the neural activations and synaptic weights respectively. $(\cdot)^{\odot 2}$ denotes element-wise square. The special case of full GP regression is obtained for $\mathbf{u} = \mathbf{f}$ and $m = n$.

Figure S2: Biological plausible neuronal network for GP regression. In the prediction phase the network outputs mean $\mathbb{E}(y)$ and variance $\mathbb{V}(y)$ of the predictive distribution for a test point $\mathbf{x}$. During learning these nodes encode the prediction errors, that are required for the synaptic updates, rather than plain predictions. The nodes are annotated with the neural activations. Plastic synapses are drawn as squiggly arrows and annotated with the synaptic anti-Hebbian learning rules. The weights of the static synapses are described in the legend. Dashed connections are only active during learning when a target value $y$ is actually present. $(\cdot)^{\odot 2}$ denotes element-wise square and $\lfloor \cdot \rfloor_+$ linear rectification, i.e. $\lfloor x \rfloor_+ = \max(x, 0)$.

# 1 GP initialized neural network

For deeper networks the quality of Dropout (and PBP) predictions increases slightly, but even then my ANN remains competitive and does not suffer from long prediction times needed to draw multiple MC samples, cf. Fig. S3.

We used the fast C++ implementation of [1] from the GitHub repo `https://github.com/HIPS/Probabilistic-Backpropagation`[1] for PBP and the updated improved code of [2] from the GitHub repo `https://github.com/yaringal/DropoutUncertaintyExps`[2] that replaced the Bayesian optimisation implementation (which was used to find hypers) with a grid-search over the hypers and used longer training times of 4,000 epochs. The evaluation was performed on a Linux workstation with Intel Xeon CPU E5-2643 v4 @ 3.40 GHz, 128 GB RAM, and NVIDIA Quadro M4000 GPU.

Figure S3: Average predictive log likelihood with Std. Errors as function of prediction time. The sparse Gaussian Process (VFE, [3]) and an artificial neural network (ANN) with architecture corresponding to a sparse GP are compared to Probabilistic Back-propagation [1] with 1 or 2 hidden layers (PBP 1, PBP 2) and Monte Carlo Dropout [2] with 1 or 2 hidden layers (Dropout 1, Dropout 2). For the latter results for 1, 10, 100, 1 000 and 10 000 MC samples are shown.

# 2 Annular water maze task

I applied the adaptation of tuning curves to data simulated to mimic a task in which rats were trained to find a hidden platform at a constant location in an annular watermaze [4], cf. Fig. S4. I modeled each location on the hidden platform as delivering the same fixed reward, while all other locations were unrewarded. In reinforcement learning [5] the expected cumulative (exponentially) discounted future reward is known as value function. Following [6], I assumed the rat imposes a GP prior over the value function. Further, values were assumed to be only available with some noise and spatially sampled according to the time in each segment. Therefore I roughly matched the number of simulated data points in each segment to the time the rat spent there in the actual experiment, cf. Fig. S4B and C. Panel D shows the true value function, which is constant on the platform and decays exponentially away from the platform due to exponential discounting of future reward. The panel also shows the fits obtained using full GP, VFE, BioNN, and BioNN with optimized place field centers. Notably, the distribution of firing fields shows the experimentally observed accumulation of place fields at the goal location [4, 7] for VFE, and even more pronounced for the BioNN with optimized place field centers.

## Footnotes

[1]commit 60ece68fe535b3b9d74cc71f996145e982872f2e

[2]commit 6eb4497628d12b0f300f4b4f6bdc386bebad565c