[Reviews · NeurIPS 2020]

Review 1

Summary and Contributions: This paper describes how Gaussian process regression could be implemented in a biologically plausible neural circuit. A key innovation of the paper is showing how not only the posterior mean but also the posterior variance can be computed. The paper makes a potentially important contribution to computational neuroscience.

Strengths: The derivation of the circuit implementation was done very carefully, as was the experimental evaluation. The results are impressive. I think the ideas here are novel and quite interesting; despite behavioral work in cognitive science suggesting that people use GPs, there has not been a plausible neural implementation. I hope it inspires experimental work to test the model predictions.

Weaknesses: If I've understood correctly, approximating the Gram matrix as an isotropic diagonal matrix (p. 4) is a drastic approximation. While this approximation is correct asymptotically, in that asymptotic regime the posterior variance goes to 0 anyways so variance doesn't matter. The mapping to specific anatomical regions (OFC, striatum, hippocampus) is very superficial. If the authors want this to be taken seriously, they need to (a) provide more compelling evidence and discussion of this mapping, and (b) describe what kind of experimental tests would be needed from neuroscientists.

Correctness: I don't see any issues with correctness.

Clarity: The paper is well written.

Relation to Prior Work: Using REINFORCE for biologically plausible updates was discussed by Seung (2003), which might be worth mentioning.

Reproducibility: Yes

Additional Feedback: abstract: "powerful flexible" -> "powerful and flexible" p. 1: I do not think the empirical results described show that the brain "performs nonlinear regression". There's no way to say whether the mapping is linear or non-linear unless you know what the input features are. p. 1: "biological plausible" -> "biologically plausible" (here and elsewhere) p. 2: "with above expression" -> "with the above expression" p. 6: "does not quite as well" -> "does not do quite as well"


Review 2

Summary and Contributions: Here it is shown that the sparse GP regression equations can be modified to be more neurological plausible, without the predictive performance suffering too much.

Strengths: The authors show the connections between a set of regression equations and functions that could potentially be implemented in a neural circuit. Table 1 shows that training the SGP from the ANN perspective can produce superior results to the alternatives tested. This is the only result that I found potentially interesting. Unfortunately, the paper is centered around the BioNN and not the ANN. Were the authors drop the biological aspect completely, focused on the ANN and showed that it is indeed a preferable alternative for regression (by comparing with SPGP and other strong baselines), I would be much more positive about the value of this work.

Weaknesses: My main concern with this work is its lack of purpose. Sure it provides a set of regression equations that are more biologically plausible that the original SGP equations. But how is this useful? How see the authors their work being leveraged in any field? From a machine learning perspective, the BioNN provided here is a _worse_ method to perform regression. So it is not of much value to people interested in regression. Surely knowing that BioNN equations are more biologically plausible is of little consolation for a lacking performance. From a neuroscience perspective, this work does not provide additional insights about how the brain works, or falsifiable hypothesis about neural function that can be tested in lab. Also, this work does not show how the brain does anything, it shows something that as of today cannot be proved not to happen, which is a fairly low standard. Even the implicit assumption of this work ("the brain performs regression, let's show how it could be doing it") is not a given. The brain does learn and predict, but might not be a universal function approximator as implied in this work, or might not be related to how GPs predict. Finally, the strong connection pushed in this work between neural regression and GPs is completely arbitrary. One could start from an RBF NN, make some tweaks and end up at the same equations obtained here. This is not an exact implementation of GPs or sparse GPs, and I would say it is just as distant from other regression techniques.

Correctness: The presented work seems correct, but the implicit claim of usefulness is dubious.

Clarity: The presentation is clear. The purpose is not. How do the authors envision their work being used?

Relation to Prior Work: Yes.

Reproducibility: Yes

Additional Feedback: Typo: "on top off" After reading the rebuttal and the other peer reviews, I acknowledge that I might be missing some point about its usefulness in neuro domain and have accordingly adjusted my confidence score, which should be enough for the paper to pass.


Review 3

Summary and Contributions: This paper devises a biologically plausible neural network that can implement Gaussian Process regression. The network can be trained by a biologically plausible learning rule (although I have some questions about this) and it can be queried at test time at input locations returning the predictive mean and marginal variance at those locations.

Strengths: The paper is well put together and clearly written. The technical contribution appears sound. The experiments are solid in as far as they go and appear promising.

Weaknesses: I have two main concerns. The first is that with a machine learning hat on, the experiments are not comprehensive enough to thoroughly evaluate the new methods. Hyper-parameter learning — critically learning the length-scale parameter of the kernel — does not appear to be tested in the larger scale experiments. Ideally there would be a wider range of datasets considered too. So assessment of the framework is incomplete from a GP / machine learning practitioner’s perspective. For me this means that the key contribution the paper makes is to the neuroscience community. On this front, I’m not sufficiently well versed with the literature to assess the magnitude of the contribution.

Correctness: Yes.

Clarity: Yes.

Relation to Prior Work: Yes.

Reproducibility: Yes

Additional Feedback: It would be useful to define the specific task setting up front — both at training and testing time — and how this relates to biology. I’m relatively happy with the test time operation: the goal is to define a neural network that takes input locations x* as inputs and returns (approximate) predictive means and variances at the queried location. I have less clarity on how the supervised training phase is handled in a biologically plausible way: in what biologically relevant scenario can the learning rule have direct access to the training outputs? Presumably these are provided by another system, such as a sensory system, but what is the biologically plausible mechanism for the learning rule to have access to them? Along similar lines, the use of the stochastic online learning rule seems to assume a setting where there are a large numbers of (x_i,y_i) training pairs which is an additional assumption that would be good to state up front. The experiments though seem to allow revisiting data points which seems biologically implausible. Longer term, if true online learning for limited data is the goal, it might be more appropriate to consider one of the online variants of the variational or FITC methods e.g. But et al. Streaming Sparse Gaussian Process Approximations, NeurIPS 2017 or Csató and Opper, Sparse online Gaussian processes,” Neural Computation, 2002. A concrete motivating example — some of which are alluded to, but not precisely specified — and a mathematical statement of the setup would really have helped me. Putting these big picture questions aside, the development of the neural networks for implementing sparse GP regression were clear and well explained. It was especially interesting that biologically plausible rules for pseudo-input optimisation were derived. The experiments are quite solid and it is interesting that the new approach can outperform FITC and VFE in some of the tested scenarios. My main reservation is that it appears that all but the last experiment set hyper-parameters (critically the length-scale) by training another method (usually VFE). So although the results are promising and interesting, they come with the caveat that the full framework has not been thoroughly demonstrated. For the Snelson data set fits shown in Fig 3A it would be illuminating to see what the error bars are doing away from the data. This dataset has a fairly uniform input distribution over the plotted range so it is hard to assess whether the method can accurately learn an input dependent predictive variance. I’d actually prefer the authors use a 1D dataset with a non-uniform input distribution (e.g. with gaps) so we get a better idea of the flexibility of the new methods. In the second set of experiments on the Snelson dataset shown in Fig. 3B and C, could you clarify how the length-scale hypers are set? Are these again taken from the relevant VFE run on each dataset? For the UCI experiments, could you clarify how the hyper-parameters were trained / set? Line 248 suggests the length-scales were taken from the VFE method — is this the case? Also, PBP and dropout methods are usually quite weak baselines and VI trained neural networks typically performs considerably better than these methods on these datasets e.g. see Bui et al. Deep Gaussian Processes for Regression using Approximate Expectation Propagation, ICML 2016. Note that this work also has FITC baselines with 50 inducing points and performs quite a lot better on Boston, Concrete, Kin8nm, Wine, Year than the results in this paper (although their results are less strong on Energy, Naval and Yacht). It might be useful to track down the source of the discrepancies. The final results using the REINFORCE method to learn hypers is promising — could you comment on the run time / computational complexity? Minor comments It’s now common to use f ~ N(\mu,\sigma^2) or p(f) = N(f; \mu,\sigma^2), but not p(f) = N(\mu,\sigma^2) as here f appears on the LHS of the equals sign but not on the right hand side which is uncomfortable. ----- Updates having read the author response and other reviews: The limitations in the paper around hyper-parameter learning are still very significant to my mind (see my review above for details). In addition, the authors’ rebuttal contains a couple of misunderstandings / misconceptions: "We thank Reviewer #3 for pointing out the superior results of deep GPs, which is, given that our ANN was derived from a standard “flat” GP, not unexpected.” The authors have misunderstood this point: the referenced paper has shallow GPs as a baseline for the deep GPs and it is these baselines which often significantly outperform the baselines in the current paper. I therefore think the statement in the rebuttal "we consider the finding that the ANN can produce superior results than VFE … on the UCI datasets” is questionable. It’s also potentially misleading as it’s taking hypers from the VFE method. "The stochastic online learning rule is used because this is the biological scenario, not because of a large number of (xi, yi) training pairs. The datasets we considered consist of fixed finite size samples from some generative process, therefore we performed multiple passes over a dataset, revisiting the same data points, cf. Fig. 2. Our update rules for the BioNN (that implements sparse GP regression) readily apply to the more biological setting where every datapoint is a new sample from the underlying generative model.” The authors need to be careful here. When performing probabilistic inference, simple ‘online learning rules’ (stochastic gradient descent and the like) will typically perform very poorly. For example, see section 2 Background and section 4 Results in Bui et al. Streaming Sparse Gaussian Process Approximations, NeurIPS 2017 for a discussion in the context of GP regression. For the reasons laid out in this paper, I would not expect that the BioNN would perform well in a realistic online setting in its current form. This limitation is not crucial, but worth clarification.


Review 4

Summary and Contributions: This manuscript introduces a novel way to approximate and learn (sparse) Gaussian Processes (GP) for regression via artificial neural networks. The main focus of the paper is towards biological plausibility, that is how a biological brain could implement Bayesian inference for regression (e.g., of state value in a maze), with GPs being a "natural" and principled choice. The authors provide formulae for setting the weights of a finite neural network (and its architecture) such that it would produce as output the GP predictive mean and variance (both for a full GP and for a sparse variational approximation), and also equations for how the weights could be learnt in a "biologically plausible" way (i.e., with local updates). Further, the authors provide simple learning rules for the GP hyperparameters and for the inducing points (for sparse GPs). Finally, the authors show empirically that their method is indeed able to learn good (sparse) GP approximations on a number of common benchmark datasets for regression, compared to other sparse GP approximations; and they also test their method in a simulated "water maze" experiment. ====== After rebuttal ======== I thank the authors for their response. I confirm my score and that this is a valuable contribution for the conference.

Strengths: Soundness: This paper is well-grounded in both theory and empirical tests, and all the claims are supported by evidence. Significance and novelty: This work is novel and timely. Its significance is potentially high, as how biological brains implement Bayesian inference is still a big open question in computational neuroscience. The link to neurophysiological data is suggestive (e.g., neuronal tuning curves resembling radial basis functions; the existence of neurons encoding mean and variance of rewards), although these empirical features can likely be linked to many different computational models. Relevance: This is a remarkable paper, tackling a very interesting problem of very high relevance for a subset of the NeurIPS community, in particular researchers at the intersection between computational neuroscience and machine learning, interested in how the brain could implement Bayesian inference. In terms of relevance, some of the results may also be of interest to the general ML community, such as ways in which finite probabilistic ANNs can be initialized from a GP (e.g., the authors show that their method is competitive with Dropout and probabilistic backprop).

Weaknesses: Novelty: While the specific implementation of GPs in neural networks is, to my knowledge, novel, there is also plenty of "related work" missing here (see below). Soundness (empirical evaluation): In addition to testing their regression method on many UCI datasets, the authors present a simple application to a water maze task. However, this scenario is barely explained, and even in the Supplementary Material not enough information is provided to fully understand the task or what the authors did. I understand that this likely depends on lack of space, and encourage the authors to expand on this section as it is in some sense the only application that connects back to a biological task.

Correctness: The claims are correct. The work includes certain approximations (e.g., to calculate the predictive variance) whose validity is then tested empirically. The empirical analysis of the method and variants on the UCI datasets against other sparse GP methods looks fair. The only "realistic" application, the watermaze task, needs more explanation to be fully evaluated.

Clarity: The paper is extremely well-written and a pleasure to read. The organization and logic of the paper and the mathematical parts are very clear and easy to follow. The only part which I found unclear (likely due to lack of space), as mentioned above, is the last results on the watermaze task (and the Supplement is also not very clear there).

Relation to Prior Work: One of the major drawbacks of the paper, which is otherwise remarkably polished, is the lack of a Related Work section (I understand there was no space in the main text, but it could have been expanded in the Supplement). In particular, there are several lines of research that are worth mentioning. For example, several other works have investigated how the brain could implement Bayesian inference, both abstractly and via neural network implementations. I appreciate that this paper differs substantially from prior work in that, as far as I know, this is the first work to suggest how the brain could perform Bayesian nonparametric regression via GPs. Still, it might be useful for the broader comp-neuro community to situate the current work in the context of other major approaches to Bayesian inference in the brain, such as "probabilistic population codes" (Ma et al., Nature Neuroscience, 2006) if relevant (or why they are not relevant). Similarly, recent work has demonstrated that simple ANNs trained on a variety of tasks reproduce Bayesian probabilistic behavior (Orhan & Ma, Nature Communications, 2017). Also, the idea that the brain might use (something akin to) GP regression is not completely novel per se and deserves some references to previous work -- although previous work in cognitive science remained abstract and did not present a neural network implementation, which is the main point of this paper. For example, see Lucas et al., Psychon Bull Rev (2015) and Wu et al., Nature Human Behaviour (2018) and references therein.

Reproducibility: No

Additional Feedback: line 59: marginal log-likelihood ==> log marginal likelihood There is no reference to released code (or plan to do so). I strongly encourage the authors to release the code to implement their method, both for reproducibility and to increase impact of their work.

[Author Response · NeurIPS 2020]

Thank you all for your helpful comments on our Comp Neuro paper. Due to space constraints we restrict attention to major comments below.

Thanks to Reviewer #1 and #4 for pointing out that behavioral work in cognitive science suggests that people indeed use GPs, and for directing us to some relevant references, Reviewer #4. We will include the behavioral Cog Sci work in the revised version of the paper and expand on it in the supplement. Missing to acknowledge this work has been to our own disadvantage as this strengthens the premise of our paper that the brain employs GPs for function learning. It is not merely a principled approach of what the brain should do, but indeed accounts for a wide variety of experimental results [Lucas et al., *Psychon Bull Rev*, 2015]. We are hopeful this dispels the doubts Reviewer #2 had regarding the basic assumption of this work.

We regard our paper primarily as a contribution to Comp Neuro, hence the focus on the BioNN as how a biological brain could implement Bayesian inference for regression. Although some of you apparently would have preferred a pure machine learning paper and greater focus on the ANN, we consider the finding that the ANN can produce superior results than VFE, PBP and MC Dropout on the UCI datasets as a surprising secondary finding, that is however of potential interest to the broader NeurIPS community. We thank Reviewer #3 for pointing out the superior results of deep GPs, which is, given that our ANN was derived from a standard "flat" GP, not unexpected. Whether a deep GP can similarly be mapped to an ANN and trained using standard deep learning techniques to further improve performance could be an interesting question for future work.

Our experiments follow the didactics used in the methods section, moving from the initial case of known hyper-parameters, to learning the noise variance $\sigma^2$ and signal strength $s$, to finally learning all hyper/variational parameters including the length scales $l$ and inducing point locations $Z$. Reviewer #3 correctly noticed, or assumed, that if the latter were not learned, they were taken from the VFE method (we will clarify this in the revised version), because we were interested in how the approximations needed to render the network biologically plausible affect its performance compared to VFE. We will test the learning of all hyper parameters in the larger scale experiments and add it to the supplement of the revised paper. If the results of Fig. 5 are indicative, this could further improve the results.

The stochastic online learning rule is used because this is the biological scenario, not because of a large number of $(x_i, y_i)$ training pairs. The datasets we considered consist of fixed finite size samples from some generative process, therefore we performed multiple passes over a dataset, revisiting the same data points, cf. Fig. 2. Our update rules for the BioNN (that implements sparse GP regression) readily apply to the more biological setting where every data point is a new sample from the underlying generative model. We will illustrate this in the revised version, either using the generative model of the Snelson data, or using a 1D dataset with non-uniform input distribution, as suggested by Reviewer #3. The supervised training phase is depicted in the somewhat busy Fig. S2. The learning rule has access to the training outputs only in the form of the prediction error that the activity of the postsynaptic neuron encodes.

While we disagree with Reviewer #2's opinion that the connection between neural regression and GPs is completely arbitrary, we agree that it is important to confirm that the approximations introduced for sake of biological plausibility still result in a BioNN that approximates the full GP well. We do so by considering the KL divergence to the full GP in Fig. 3 and Table 2, which shows that our BioNN approximates the full GP not quite as well as VFE, the starting point of BioNN's derivation, but better than FITC.

We appreciate that Reviewer #4 values the application to a water maze task, the biological task that due to lack of space did not make it into the main text. We will greatly expand on this section in the revised version to provide more information and better explanation.

For reproducibility and potentially increased impact of out work, we will release the code to reproduce all figures and tables of the paper. We actually have cleaned up the code for public release already, but failed to do so in time for the supplementary material deadline.

[Meta-Review · NeurIPS 2020]

This paper presents a biologically plausible construction of Gaussian process regression. The 4 reviewers were split into two camps (two strong accepts and two rejects), where one argued that the paper was an exciting and significant contribution to computational neuroscience and the other arguing that the GP construction and empirical evaluation were insufficient for an ML paper. There was extensive discussion, with the ML camp agreeing that they wouldn't argue strongly against acceptance if the work is indeed interesting to computational neuroscience. As NeurIPS includes computational neuroscience as a focus area and the reviewers focusing on that aspect found the work very exciting, it would seem this paper could be quite interesting to researchers in that sub-community.